# Antimycobacterial and Anticancer Properties of *Myrtus communis* Leaf Extract

**DOI:** 10.3390/ph17070872

**Published:** 2024-07-02

**Authors:** Mushtaq Ahmad Mir, Lamis Ahmad Memish, Serag Eldin Elbehairi, Nasreena Bashir, Faris Saif Masoud, Ali A. Shati, Mohammad Y. Alfaifi, Ahmad M. Alamri, Sultan Ahmad Alkahtani, Irfan Ahmad

**Affiliations:** 1Department of Clinical Laboratory Sciences, College of Applied Medical Sciences, King Khalid University, P.O. Box 3665, Abha 61421, Saudi Arabia; 2Biology Department, Faculty of Science, King Khalid University, Abha 9004, Saudi Arabia; 3Microbiology Laboratories, Southern Region Armed Forces Hospital, Khamis Mushayet 62413, Saudi Arabia

**Keywords:** *Myrtus communis*, anticancer, apoptosis, antibacterial, biofilm

## Abstract

Background: Plant-derived products or extracts are widely used in folk/traditional medicine to treat several infections, ailments, or disorders. A well-known medicinal herb, *Myrtus communis* is an evergreen fragrant plant native to the Mediterranean region that has been used for ages in traditional medicine around the world. Materials and methods: The microplate alamarBlue assay and the well diffusion method were used to evaluate the zone of inhibition and MIC, respectively. The double-disc diffusion method was used to investigate the synergy between antibiotics and the extract. The crystal violet method was used to investigate biofilm development. The SulphoRhodamine-B assay and DNA flow cytometry were used to investigate the proliferation and subsequent distribution of cells among different phases of the cell cycle. The apoptotic and necrotic phases of the cancer cells were examined using flow cytometry in conjunction with Annexin V-FITC/PI labeling. Using the IBM SPSS statistical program, a one-way ANOVA with Tukey’s post hoc test was employed for statistical analysis. Results: The ethanolic leaf extract of *M. communis* showed a strong growth inhibition effect (zone of inhibition: 20.3 ± 1.1–26.3 ± 2.5 mm, MIC: 4.88–312.5 µg/mL, and MBC: 39.07–1250 μg/mL) against several rapidly growing and slow-growing mycobacterial strains in a dose-dependent manner. Damage to the cell wall of bacterial cells was determined to be the cause of the antimycobacterial action. The extract inhibited biofilm formation (MBIC of 9.7 µg/mL) and eradicated already-formed mature and ultra-mature biofilms of *M. smegmatis*, with MBEC values of 78 µg/mL and 156 µg/mL, respectively. Additionally, the extract exhibited potent anticancer effects against diverse cancer cell lines of the breast (MCF-7), liver (HepG2), cervix (HeLa), and colon (HCT116) (IC_50_ for HCT116: 83 ± 2.5, HepG2: 53.3 ± 0.6, MCF-7: 41.5 ± 0.6, and HeLa: 33.3 ± 3.6) by apoptosis after arresting the cells in the G_1_ phase of the cell cycle. Conclusions: These results suggest that *M. communis* leaf extract is a potential source of secondary metabolites that could be further developed as potential anticancer and antimycobacterial agents to treat diverse types of cancers and mycobacterial infections.

## 1. Introduction

*Myrtus communis* or Myrtle (Arabic name: Aas or Hadas), belonging to the family Myrtaceae (Saudi Arabia refers it as Mesk Al-Madena), is a notable therapeutic herb. It is an evergreen aromatic and medicinal plant found in the Mediterranean region, along with other nations, such as Iraq, Jordan, the southern and eastern provinces of Saudi Arabia [1,2], and Iran, in the Middle East. The various parts of this herb, such as its berries, leaves, and fruits, have been extensively used as folk medicine for several centuries. The herb is traditionally used for the treatment of disorders such as inflammation, diarrhea, hemorrhoids, peptic ulcers, pulmonary diseases, and skin diseases. It possesses a broad spectrum of pharmacological and therapeutic effects, such as antidiabetic, antioxidative, antiviral, antibacterial, antifungal, anticancer, hepatoprotective, and neuroprotective activity [1]. While the ethanolic leaf extract of *M. communis* demonstrates a range of biological activities, including antinociceptive, antidiabetic, anti-inflammatory, and antioxidant properties [2,3,4], little is known about the extract’s antibacterial properties, particularly its antimycobacterial properties. Not much progress has been made in understanding the anticancer activity of *M. communis*, except in a few studies, wherein mostly essential oils, chloroform, hexane, ethyl acetate, and methanol extracts have been screened for anticancer activity [1,5]. 

Due to the emergence of drug resistance and the reduced rate of new antibacterial agent development, the treatment of bacterial infections is strained. Biofilms, one of the causes of drug-resistant tuberculosis [6], are surface-attached microbial communities in which microbial cells are entrenched in extracellular polymeric substances (EPSs) formed by the cells themselves [7]. As a result, biofilms serve as effective physical barriers to antibiotic and nutrient penetration. Therefore, there is a need to develop new drugs to treat bacterial infections, for which natural compounds are appropriate.

Mycobacteria, including *M. tuberculosis* and other non-tuberculosis mycobacteria (NTM) species, are strong biofilm producers [8], and they cause several types of diseases in humans. Tuberculosis, caused by *M. tuberculosis*, is a leading cause of 1.5 million deaths each year and a major contributor to antimicrobial resistance. Non-tuberculosis mycobacteria (NTM), once thought to be harmless environmental saprophytes and dangerous to immunocompromised and lung-defective individuals, are now causing a spectrum of diseases that include tuberculosis (TB)-like pulmonary and extrapulmonary disease, visceral and disseminated disease, and cervical lymphadenitis in immunocompetent individuals. In Saudi Arabia and other Gulf countries, *M. fortuitum* and *M. abscessus* were found to comprise approximately 36% and 21% of the clinical isolates of mycobacterium [9].

Cancer, also called malignancy, caused nearly ten million deaths in 2020 [10]. Breast cancer is one of the leading causes of morbidity and mortality for women worldwide [11]. Hepatocellular carcinoma (HCC), the most common primary malignancy of the liver, is the leading cause of death in people with cirrhosis [12]. The deregulation of apoptosis, a biological phenomenon regulating tissue development and maintaining homeostasis [13], is one of the hallmarks of cancer. Moreover, various epidemiological studies have substantiated the link between infection and cancer development [14]. Using multidisciplinary surgical and non-surgical approaches, including systematic chemotherapy, significant progress has been made in the diagnosis and treatment of cancer. Because of the undesirable side effects and efficacy of conventional chemotherapy, a variety of cytotoxic drugs are incompetent. Therefore, to avoid the onset and prognosis of infectious diseases and non-infectious diseases like cancer, a novel drug with both antibacterial and anticancer characteristics with no unfavorable side effects is required.

In the current study, the ethanolic leaf extract of *M. communis* was examined for its antibacterial properties against a variety of mycobacterial strains, including *M. tuberculosis* and non-tuberculosis mycobacterial strains. Additionally, we looked at the effects of *M. communis* extract on the development of biofilms in *S. aureus* and *M. smegmatis*, the latter being used as a model organism for *M. tuberculosis* pathogenesis. Moreover, the ethanolic leaf extract was examined for its impact on cell viability, cell cycle arrest, and the apoptosis of four different cancer cell lines. This is the first report of the ethanolic leaf extract of *M. communis* exhibiting anticancer efficacy against cancer cell lines such as MCF-7, HepG2, HCT-116, and HeLa.

## 2. Results

### 2.1. M. communis Leaf Extract Strongly Inhibited the Growth of Mycobacterial Strains

To determine the antimycobacterial activity of the *M. communis* leaf extract, all ten mycobacterial strains were uniformly streaked individually on TSA plates. The zone of inhibition of growth by the extract for each strain was determined using the well diffusion method. The mycobacterial strains sensitive to the extract produced a clear zone of no bacterial growth around the well (Figure 1). The zone of inhibition ranged from 20.3 ± 1.1 mm to 26.3 ± 2.5 mm (Table 1), suggesting that all the mycobacterial strains, including tuberculosis and non-tuberculosis (SGM and RGM), were strongly inhibited by the extract.

The MIC of *M. communis* extract against each mycobacterial strain was determined by the microplate alamarBlue assay. AlamarBlue, a growth indicator, changes color from blue to pink during the growth of cells. The bacterial strains were treated with 2-fold dilutions of *M. communis* leaf extract ranging from 0.02 g/mL to 3 µg/mL in a 96-well plate. The lowest concentration of the extract at which no change in color from blue to pink occurs is considered the MIC. As shown in Figure 1, 4.8 µg/mL is the MIC for *M. kansasii* ATCC35775. The MIC of the extract against each bacterial strain was determined, and the results are presented in Table 1. It is clear from the table that MIC ranged from 4.8 µg/mL to 312.5 µg/mL. Only two mycobacterial strains, *M. fortuitum ATCC6841* (MIC: 156.2 µg/mL) and *M. tuberculosis* RIF-R (MIC: 312.5 µg/mL), were comparatively less sensitive to the extract.

From the alamarBlue assay plate, the cultures of each strain from the wells having extract concentrations above the MIC were streaked on TSA plates. The concentration of the extract at which no visible growth of bacterial colonies was detected for a particular strain was considered the MBC for said strain. The MBC ranged from 39.07 µg/mL to 1250 µg/mL for all mycobacterial strains tested (Table 1). It was found that there was a consistent correlation between MIC and MBC for each mycobacterial strain tested. The strains with a lower MIC showed a lower MBC, and vice versa. Consistent with the MIC trend, *M. fortuitum ATCC6841* (MBC: 937 µg/mL) and *M. tuberculosis* RIF-R (MBC: 1250 µg/mL) showed comparatively higher MBC values.

### 2.2. Growth Kinetics of the Bacterial Strains in the Presence of the Extract

MABA, an adequate method for testing antimicrobial activity, was used to monitor the growth of extract-susceptible bacterial strains in the continuous presence of different concentrations of the extract. AlamarBlue, a growth indicator, is a fluorescent dye that changes its color from blue to pink in response to the reduced environment of the growing cells. In our study, 0.5 McFarland bacterial cultures diluted a hundred times in trypticase soy broth containing 1× alamarBlue were incubated with the extract at its MIC, 2×MIC, and 4×MIC. For a control, the cultures were incubated with ethanol. Growth media containing 1× alamarBlue were used as a blank for their sterility. Growth, at regular intervals of 30 min for *S. aureus* and 1 hour for *M. smegmatis*, was monitored using the FLUOstar^®^ Omega microplate reader (BMG Labtech, 77799 Ortenberg, Germany) over time. Blank-corrected average fluorescence units (Ex_560_/Em_590_) plotted against the time for each strain (Figure 2) showed that both strains tested had a normal pattern of growth in the presence of ethanol. However, a decrease in growth was observed for both the two strains tested, with a corresponding increase in the concentration of extract from the MIC to 4×MIC. These results suggest that the growth-inhibiting effect is dose-dependent.

### 2.3. M. Communis Leaf Extract Damages the Bacterial Cell Wall

To investigate the effect of the extract on cellular morphology, *M. smegmatis* cells were treated with the extract. For a control, the cells were treated with the carrier solvent ethanol under similar conditions. After fixation and negative staining, the cells were observed under a transmission electron microscope. The tomographic images demonstrated that the extract-treated cells have a damaged cell boundary compared to the control cells (Figure 3). This suggests that the extract targeted the cell wall, which is consistent with our earlier observation [15].

To further explore the cell wall-damaging effect of the extract, the *M. smegmatis* cells in the presence of the extract were treated with several commercial antibiotics, including cell wall-targeting vancomycin and bacitracin. Antibiotic discs such as gentamycin, tetracycline, ofloxacin, vancomycin, chloramphenicol, polymyxin B, colistin, and bacitracin were impregnated with the extract and tested for their zones of inhibition. For a control, antibiotic discs were impregnated with ethanol and tested under similar conditions. It was found that the zone of inhibition of vancomycin, colistin, bacitracin, and chloramphenicol in the presence of the extract was ≥5 nm higher than that of the antibiotic alone. However, no change in the size of the inhibition zone was observed for polymyxin B. On the contrary, the extract exhibited a negative effect on gentamycin, clarithromycin, ofloxacin, and tetracycline (Table 2).

To further explore the interaction between the extract and antibiotics, the double-disc diffusion method was employed, wherein discs of antibiotic and extract were kept at the half sum of their inhibition zone sizes. As shown in Figure 4, clear bridging was observed between the inhibition zones of vancomycin, bacitracin, and the extract. However, no such bridging was observed for the remaining antibiotics tested (Figure 4 and Appendix A). Bacitracin and vancomycin target the cell wall of the bacteria by blocking the synthesis of the cell wall component peptidoglycan, which is abundant in *Mycobacteria*. The synergistic action between the extract and the antibiotics vancomycin and bacitracin, as well as the scarring of the bacterial cell walls, as shown by SEM, indicated that the extract’s bio-actives targeted the cell walls of the *M. smegmatis* cells.

### 2.4. M. communis Leaf Extract Inhibits the Biofilm Formation of M. smegmatis and S. aureus

Mycobacterial strains are believed to form a biofilm [16], which is the main barrier to antimicrobials killing the resident microorganism. We attempted to investigate whether *M. communis* leaf extract could affect the biofilm formation of mycobacterial strains. *M. smegmatis* was chosen as an ideal model microorganism because it is a non-pathogenic and fast-growing microorganism that shares many similarities with *M. tuberculosis*. The results obtained for the mycobacterial strains will be described later, as they are not the focus of the present study. For a control, we explored the effect of the extract on *S. aureus*, which is a robust biofilm-forming bacterium.

Before investigating the effect of the extract on biofilm formation, the MIC of the extract for both strains was determined by the microdilution method. In brief, 0.5 McFarland cultures of *S. aureus* and *M. smegmatis* were individually treated with 2-fold diluted *M. communis* leaf extract, ranging from a final concentration of 10 mg/mL to 0.6 µg/mL. The MIC of the extract for *M. smegmatis* and *S. aureus* was found to be 39 µg/mL and 19.4 µg/mL, respectively. To study the effect of the extract on biofilm formation, *S. aureus* and *M. smegmatis* strains were individually treated with increasing 2-fold *M. communis* extract concentrations ranging from 4.85 µg/mL (1/4MIC) to 77.6 µg/mL (4×MIC) and 9.75 µg/mL (1/4MIC) to 156 µg/mL (4×MIC), respectively. Subsequently, the biofilms formed were developed by the crystal violet staining method. The percentage of biofilm inhibition calculated at each concentration of extract for *S. aureus* (Figure 5A) and *M. smegmatis* (Figure 5B) was plotted. It was found that the biofilm formation of *S. aureus and M. smegmatis* was inhibited by 51% and 70% using 9.7 µg/mL of the extract, respectively. Therefore, the minimum biofilm inhibition concentration (MBIC) of the extract for both strains was considered to be 9.7 µg/mL. The MBIC was defined as the minimum concentration of the extract that exhibits the highest level of biofilm inhibition without affecting growth.

We further evaluated the effect of the extract on already-formed mature and extra-mature biofilms. For mature biofilm formation, the bacterial cultures of *S. aureus* and *M. smegmatis* were incubated at 37 °C under static conditions for 24 and 48 h, respectively. For extra-mature biofilm formation, the *S. aureus* and *M. smegmatis* cells were incubated under similar conditions for 48 and 72 h, respectively. The biofilms formed were washed off the planktonic cells and further treated with the extract for 24 h. Biofilms retained after extract treatment were stained and quantitated by the crystal violet method. Under similar conditions, the biofilms formed in the control wells were treated with ethanol only and subsequently stained and quantitated by the crystal violet method. It is clear from Figure 6A,B that the mature and extra-mature biofilms in *M. smegmatis* were eradicated by 100% and 90% at 78 µg/mL and 156 µg/mL, respectively. Therefore, the minimum biofilm eradication concentration (MBEC) values of the extract for mature and extra-mature biofilms of *M. smegmatis* were considered to be 78 µg/mL and 156 µg/mL, respectively. Contrarily, for *S. aureus*, the MBEC for mature and extra-mature biofilms was found to be 77.6 µg/mL. The MBEC was defined as the minimum concentration of the extract that exhibits the greatest eradication of already-formed biofilms. At the respective MICs of the extract for *M. smegmatis* and *S. aureus*, 80–88% and 35–45% of their already-formed biofilms were eradicated, respectively. These results suggested that the mature biofilms of *M. smegmatis* were comparatively more susceptible to the extract.

### 2.5. M. communis Leaf Extract Inhibited the Proliferation of Cancer Cell Lines

To investigate the effect of *M. communis* leaf extract on the proliferation of different types of cancer cells, viz. breast, liver, colon, and cervix cancer cells, an SRB assay was carried out. The cell toxicity of the treatment with different concentrations of extract for 72 h was measured by the percentage viability of the cancer cells.

The results showed that the viability of the cells decreased in a dose-dependent manner (Figure 7), killing 75% of all cells at a concentration of 100 μg/mL. A gradual increase in the concentration of the extract resulted in a gradual increase in the growth inhibition of all four types of tumor cells upon their independent treatment with the extract. The cytotoxic activity of the extract was strong for HeLa cells (IC_50_: 33 μg/mL) (Table 3), while for other cells, it was moderate (IC_50_ for HCT116: 83 ± 2.5, HepG2: 53.3 ± 0.6, and MCF-7: 41.5 ± 0.6).

### 2.6. M. communis Leaf Extract Caused Cell Cycle Arrest and the Induction of Apoptosis in Cancer Cells

One of the major causes of the inhibition of cellular growth is cell cycle arrest. To determine whether the growth inhibition of cancer cell lines was due to cell cycle arrest in a particular phase, the cell lines were treated individually with the ethanolic leaf extract at their IC_50_ concentrations. After treatment, the cell cycle profile of each cell line was determined by PI staining, followed by DNA flow cytometry analysis. Table 4 and Figure 8 show that a significant proportion (~10% more than the control) of all the tumor cell lines was arrested in the G_1_ phase of the cell cycle, with a corresponding drop in the percentage of cells in the S-phase upon treatment with the ethanolic leaf extract of *M. communis.* Moreover, a marked increase in the percentage of cells in the G_2_/M phase (3–5% more than the control) was observed for all cancer cell types except HCT116. These results demonstrate that the extract arrested all types of cancer cells in the G_1_ phase of the cell cycle, though the percentage of cells showed a variation.

To further investigate the extract-induced inhibitory effect, cells treated with the extract were analyzed in a flow cytometer after Annexin V-FITC/PI staining. Analysis of the percentage of cells detected in different stages of apoptosis (Figure 9, Appendix A) indicated that the major cellular populations of the MCF-7 (97.15%) and HeLa (97.38%) cell lines were in the late apoptotic stage, while 71.32% of the HCT-116 cells were found to be in the early stage of apoptosis. However, significant proportions of HepG2 cells were found in both the early (50.6%) and late (47.8%) stages of apoptosis. Less than 1% of all cancer cells were found in the necrotic stage of apoptosis, apart from for the HCT-116 cells, a significant proportion of which (17.66%) were found to be in the necrotic stage of apoptosis. The cells treated with the extract were further analyzed under a fluorescence microscope for nuclear morphological changes (apoptosis or necrosis) after acridine orange/ethidium bromide staining. As usual, the major hallmarks of apoptotic cell death are DNA fragmentation and a loss of membrane asymmetry. The extract of *M. communis* induced morphological changes, DNA fragmentation, nuclear shrinking, etc., which are characteristics of various stages of apoptosis, viz. early- or late-phase apoptosis or necrosis (Appendix A). Altogether, these results suggest that the major cellular populations of all four tumor cell lines tested were in the apoptotic stage upon treatment with the leaf extract.

## 3. Discussion

Although substantial improvements have been made in medical technology, there is no cure for almost any cancer around the globe. In addition to non-communicable diseases like cancer, communicable diseases are an additional burden on human health and the economy. Failure to the treat the diseases caused by *M. tuberculosis* and NTM with available commercial antibiotics is a growing concern. The thick cell wall and dormant ability of these bacterial species demand the development of new therapeutic agents.

Natural products extracted in crude or pure form and used for the treatment of various ailments were obtained from medicinal plants. Though herbal medicine has some advantages over a purified constituent compound [17], there is a growing global demand to identify and purify the constituent effectively.

The other majorly significant aspect of this study is the extract’s strong growth-inhibiting effect on several NTM strains, including slow-growing and rapidly growing strains (Table 1). The present study is the first to report that *M. communis* leaf extract inhibited the growth of *M. tuberculosis* and several NTM strains, which is consistent with the growth-inhibiting effect of *M. communis* leaf essential oil against several strains of *M. tuberculosis* [18]. These results suggest that there is the potential to use the extract for the treatment of tuberculosis and non-tuberculosis diseases. *M. kansasii,* known to cause pulmonary disease in immunocompromised individuals or those with underlying pulmonary diseases such as silicosis, was strongly inhibited by *M. communis* leaf extract. Among all the bacterial strains tested, *M. communis* extract exhibited the highest zone of inhibition and lowest MIC against *M. kansasii* ATCC35775 (Table 1). *M. abscessus* and *M. fortuitum* were significantly inhibited, while *M. xenopi* showed marked susceptibility to *M. communis* extract. Surprisingly, *M. tuberculosis* strains, especially the rifampin-resistant *M. tuberculosis* strain, showed less sensitivity to the extract (Table 1). These findings corroborate various earlier studies wherein pyrogallol, Linalool, 1, 8-cineole, α-terpineol, linalyl acetate, D-limonene, and β-caryophyllene, the major constituents of the extract (Appendix A), exhibited not only antibacterial and antibiofilm activities against Gram-positive and Gram-negative bacteria, but anticancer activities as well [19,20,21,22,23,24,25]. More interestingly, it was reported earlier that the three constituents of *M. communis* essential oil—limonene, 1,8-cineole, and α-pinene—exhibited antimycobacterial activities against four strains, namely *M. tuberculosis*, *H37Rv*, *H37Ra*, and *M. paratuberculosis* [18].

According to our previous study [15], we believe that the cell wall of the susceptible bacterial cells was the probable target for the action of the extract, because the tomographic images of extract-treated *M. smegmatis* cells (Figure 3) showed damage to the cell wall. This could be one of the leading causes of cell content leakage and the subsequent death of bacterial cells. In addition, the bridging of the inhibition zones of antimicrobials demonstrates their synergistic action [26]. We observed that the antibacterial activity of the extract was synergistic in combination with the cell wall-targeting antibiotics bacitracin and vancomycin (Figure 4), further indicating that the cell wall was the target for the action of the extract. These observations corroborate reports describing the rupturing of the cell walls of *E. coli* and *Staphylococcus aureus* cells treated with α-terpineol and limonene, respectively [23,27]. Biofilms are one of the impediments to the access of antimicrobials for the treatment of bacterial infections. Biofilms of *S. aureus* are often associated with chronic infections and infected embedded medical devices. Phytochemicals inhibiting the formation of biofilms in *S. aureus* have been studied to a certain extent. For mycobacteria, including NTM and *M. smegmatis,* the extracts and essential oils of a few medicinal plants have been investigated for the inhibition of biofilm formation [16,28,29]. In this study, we attempted to investigate the effect of ethanolic *M. communis* leaf extract on the biofilm formation and eradication of already-formed biofilms of *M. smegmatis* and *S. aureus* (Figure 5 and Figure 6). *S. aureus*, a pathogen commonly seen in clinical and laboratory settings, and *M. sm*egmatis, a model organism for studying the pathogenesis of *M. tuberculosis* and NTM, were included in the biofilm assays. Our results suggest that the *M. communis* extract exhibited strong biofilm inhibition, and it eradicated already-formed biofilms of both *M. smegmatis* and *S. aureus*. The biofilm inhibition could be attributed to the major constituents of the extract (Appendix A), as several of them possess antibiofilm activities [20].

In addition to antibacterial activity, the ethanol extract showed strong cytotoxic activity against all of the four cancerous cell lines tested (IC_50_: 33.3 ± 3.6–83 ± 2.5 μg/mL). All types of cancer cells were arrested in the G_1_ and G_2_/M phases of the cell cycle, except for the HepG2 and HeLa cell lines, wherein the population of cells decreased in the G_2_/M phase of the cell cycle. It has been shown in previous studies that a sequence of events occurs as the damaged cells proceed through their arrest into the G_1_ or G_2_/M phases. Eventually, after passing through aberrant mitosis, they undergo apoptosis. Several distinct pathways lead to apoptosis, which usually occurs as a later event in cell death. As shown in Figure 9, a significant population of cells was found in the early and late phases of apoptosis. This is in agreement with the fact that anticancer molecules arrest cells in the growth phase of the cell cycle and subsequently induce apoptosis, which leads to cell death [30].

It is believed that bacterial infection and dysbiosis lead to the development of cancer. For example, *Fusobacterium nucleatum* has been implicated in various types of cancers, including colorectal cancer, esophageal cancer, and gastric cancer [31]. Similarly, *H. pylori* has a role in gastric tumorigenesis [32]. Dysbiosis, a compositional and functional alteration of the gut microbiome, also contributes to the development of various pathological conditions, including cancer. Therefore, the antibacterial and anticancer properties of the extract need to be investigated in a mouse model of infection. We envisage its use in treating mycobacterial infections and possibly being preventive of lung cancers in chronic TB patients, which require further in vivo studies of the bioavailability, stability, and cytotoxicity of normal flora and host cells. Although, the epidemiological link between chronic tuberculosis and lung cancer substantiates *Mtb* infection as a risk factor for lung cancer [33]. The present study suggests that the extract contains compounds that not only inhibit the growth of infectious bacterial strains but also exhibit anticancer activity in vitro. However, it needs to be examined whether the extract has any effect on normal host flora.

## 4. Materials and Methods

### 4.1. Preparation of Plant Extract

Leaves of the medicinal plant *M. communis*, native to Faifa Mountain, located in the east of Jazan, Saudi Arabia, were collected in the months of summer and sun-dried. The specimen has already been identified and confirmed [15]. Dried leaves of the native medicinal plant *M. communis* were ground into a fine powder. The extract was prepared in ethanol by the Soxhlet extraction method. Fifty grams of the leaf powder was incubated with 200 mL of absolute ethanol for 2 h in the Soxhlet extractor. The extract was filtered through Whatman-1 paper, and the filtrate subsequently obtained was poured into Petri dishes. The Petri dishes were left open at room temperature until the complete evaporation of ethanol. The yield of the dried extract was 15.3% of the dried powdered leaves. The dried extract of *M. communis* was redissolved either in ethanol or DMSO at the desired concentrations and used in assays.

### 4.2. Bacterial Strains and Media

Ten mycobacterial strains, including laboratory and reference strains, viz. *M. abscessus*, *M. kansasii*, *M. mucogenicum*, *M. xenopi*, *M. tuberculosis* Rif-R, *M. kansasii* ATCC35775, *M. tuberculosis* ATCC25177/H37Ra, *M. avium* ATCC25291, and *M. fortuitum* ATCC6841, were grown in Middlebrook 7H9 broth supplemented with OADC. The mycobacterial strains, including both slow-growing (SGM) and rapidly growing mycobacterial (RGM) strains, were maintained in the TB laboratory (BSL3 facility) at Southern Region Military Hospital, Khamis Mushait, Saudi Arabia. Trypticase soy broth (TSB) supplemented with 2% glucose was used for *M. smegmatis* and *S. aureus* in their growth assays and other assays, such as the TEM, biofilm inhibition, and antibacterial assays. Antibiotic discs, polymyxin B (50 mg) and bacitracin (130 mg), were obtained from Bio-Rad. Discs of vancomycin (30 mg) and clarithromycin (15 mg) were purchased from Oxoid Ltd., Hampshire, England, while discs of colistin (10 mg), gentamicin (10 mg), ofloxacin (5 mg), chloramphenicol (30 mg), and tetracycline (30 mg) were purchased from Abtek Biologicals Ltd., Liverpool, UK. The CLSI’s standards for antimicrobial susceptibility testing were followed [34].

### 4.3. Zone of Inhibition

The agar well diffusion method is widely used to evaluate the antimicrobial activity of plant extracts. The zone of inhibition was determined as described in [35], with slight modifications. Wells with a 6 mm diameter were punched aseptically on TSA plates with a sterile tip. Using aseptic techniques, inoculums of 0.5 McFarland were prepared in normal saline. Subsequently, the bacterial suspension was diluted 100-fold in OADC-supplemented Middlebrook 7H9 growth media. The diluted bacterial suspensions were spread over the agar plate’s surface using a sterile cotton swab to obtain uniform microbial growth on the plates. Then, 20 μL of 0.4 g/mL of *M. communis* extract was dispensed into the wells of the plate, while, for a control, 20 μL of ethanol was dispensed into separate wells. Depending on the growth status of a bacterial strain, the plates were incubated aerobically for 4 days to 4 weeks at 37 °C. A ruler was used to measure the diameter (mm) of a clear zone around the well. The actual zone of inhibition was determined using the following equation: zone of inhibition = (average diameter of the zone of inhibition by extract—average diameter of the zone of inhibition by ethanol) + the diameter of a well (6 mm). For the double-disc diffusion method, the antibiotic and the extract discs were kept at the half sum of their zones of inhibition. The bridging between the zones of inhibition of the antibiotic and extract indicates their synergy.

### 4.4. Minimum Inhibitory Concentration (MIC)

MIC was determined by the broth dilution method as described in [35], with slight modifications. In brief, 0.5 McFarland bacterial cultures in normal saline were diluted 100-fold in OADC-supplemented Middlebrook 7H9 media containing 1× alamarBlue. Next, 190 μL of diluted culture was dispensed into the wells of a 96-well plate, with the peripheral wells being filled with 200 μL of distilled water to avoid evaporation during the incubation period. The 11th column was filled with 200 μL of OADC-supplemented Middlebrook 7H9 media to control the sterility of the growth medium. Wells of the column were reserved for a control, to which 10 μL of ethanol was added. For each bacterial strain, three wells per dilution of extract were used. The plates were incubated for 4 days to 4 weeks. The lowest concentration of the compound at which no visible growth (no color change) was observed was considered the MIC for the bacterial strain tested. For *M. smegmatis* and *S. aureus*, the cultures grown in TSB (trypticase soy broth) were supplemented with 2% glucose.

### 4.5. Minimum Bactericidal Concentration

The minimum bactericidal concentration (MBC) is the lowest concentration of an antimicrobial at which a microorganism is completely killed. A sterilized loop was used to streak the cultures from the wells of the MIC up to the wells of the highest concentration of the extract on TSA plates. The extract concentration where no growth was detected was considered the MBC for the bacterial strain tested.

### 4.6. Microplate AlamarBlue Assay (MABA)

The microplate alamarBlue^®^ assay (MABA) is a sensitive, rapid, and nonradiometric method that evaluates cellular health and offers the potential for screening large numbers of antimicrobial compounds. MABA was performed as described in [35], with a slight modification. In brief, 190 μL of the 100-fold diluted culture of a bacterial strain in 2% glucose-supplemented TSB containing 1× alamarBlue was dispensed into the wells of a 96-well plate. Then, 10 μL of the extract at its MIC, 2×MIC, and 4×MIC was dispensed into triplicate culture wells. For a positive control and medium sterility, 10 μL of ethanol was added to triplicate wells of culture and growth media, respectively. The peripheral wells were filled with 200 μL of sterile water to avoid evaporation of the well content. The plate was sealed with a breathable sealing film (Merck) to allow for the exchange of gases. The plate was incubated in a FLUOstarr^®^ Omega microplate reader. Fluorescence readings (excitation/emission maxima at 544/590 nm) were recorded every 30 minutes and 1 hour for *S. aureus* and *M. smegmatis*, respectively. The average values of blank-corrected fluorescence units were plotted against time using Microsoft Excel software. Standard deviations were calculated, and graphs were constructed using Microsoft Excel software, version 16.66.1.

### 4.7. Crystal Violet Biofilm Assay

A few bacterial colonies of *S. aureus* and *M. smegmatis* mc^2^155 grown on TSA plates were suspended in normal saline to obtain 0.5 McFarland (10^8^ CFU/mL) bacterial suspensions, which were later used in the biofilm assay, carried out as described in [36]. In brief, the above-prepared 0.5 McFarland bacterial suspensions were diluted 100 times in TSB supplemented with 2% glucose, and 190 µL of the diluted cultures was dispensed into the wells of a 96-well plate. Then, 10 µL of *M. communis* extract with a 2-fold increase in final concentration ranging from 1/2MIC to 4×MIC for each bacterial strain was dispensed into triplicate wells. For the positive control, 10 µL of absolute ethanol, a carrier solvent, was similarly dispensed into the triplicate culture wells for each bacterial strain. As a control for the binding of crystal violet to the walls of the wells, 200 µL of sterile TSB was dispensed into triplicate wells. The plates were incubated for either 24 h (for *S. aureus*) or 48 h (for *M. smegmatis*) at 37 °C under static conditions. Biofilms were developed as described by Christensen et al. [37] and measured at an absorbance of 570 nm using a *FLUOstar*^®^
*Omega* microplate reader featuring BMG LABTECH’s proprietary tandem technology. The percentage of biofilm inhibition was calculated using the following formula: “percentage of biofilm inhibition = (1 − average OD_570_ of sample/average OD_570_ of control) × 100”. The percentage of biofilm inhibition was plotted against the extract concentration using Microsoft Excel software, version 16.66.1. The standard deviations (±SD) shown by the error bars were plotted onto a graph.

### 4.8. Transmission Electron Microscopy (TEM)

The *M. smegmatis* cells were treated with 10×MIC of extract for 12 h at 37 °C in a shaking incubator at 200 rpm. To rule out the effect of the carrier solvent, ethanol, on cell morphology, the final concentration of ethanol was maintained at a minimum concentration of 1.6%. For a control, the cells were treated with the same concentration of 1.6% ethanol under similar conditions.

Both control and extract-treated *M. smegmatis* cells were harvested at 5000× *g*. The cell pellets were immediately resuspended in 2.5% paraformaldehyde–glutaraldehyde fixative for 2 hours at 4 °C. The samples were further centrifuged at 5000× *g*, and the pellets were washed twice with sodium cacodylate buffer. The samples were adsorbed for 1 min onto a formvar-coated 200-mesh copper grid. The adsorbed samples were washed two times in sodium cacodylate buffer, and negative staining was carried out by immersing coated grids for 20 seconds in 1% uranyl acetate, followed by destaining in sterile Milli-Q water and air-drying. Imaging was performed at 80 Kv using a JEM-1011 transmission electron microscope (JEOL Co., Tokyo, Japan) in the EM-Unit, College of Medicine, King Khalid University. TEM images were recorded at 15,000×magnification.

### 4.9. Cell Culture

Human hepatocellular carcinoma cells (HepG2), colorectal adenocarcinoma cells (HCT116), breast adenocarcinoma cells (MCF-7), and human cervix adenocarcinoma cells (HeLa) were obtained from the American Type Culture Collection (ATCC). The cells were maintained in RPMI-1640 supplemented with 100 μg/mL penicillin and heat-inactivated fetal bovine serum (10% *v/v*) in a humidified 5% (*v/v*) CO_2_ atmosphere at 37 °C.

### 4.10. Cytotoxicity Assessment

The cytotoxicity of the extract was tested against human tumor cells using the Sulphorhodamine B assay (SRB). Healthy growing cells were cultured in a 96-well tissue culture plate (3000 cells/well) for 24 h before being treated with the extract to allow for the attachment of the cells to the plate. The cells were exposed to five different concentrations of the extract (0.01, 0.1, 1, 10, 100, and 1000 μg/mL) and dissolved in DMSO. For a control, the cells were treated with DMSO alone. Triplicate wells were included for each extract concentration. The plate was incubated for 72 h at 30 °C and subsequently fixed with TCA (10% *w/v*) for one hour at 4 °C. After several washings, the cells were stained with 0.4% (*w/v*) SRB solution for 10 min in the dark. The excess stain was washed with 1% (*v/v*) glacial acetic acid. After drying overnight, the SRB-stained cells were dissolved in tris–HCl, and absorbance was measured using a microplate reader at 540 nm. The linear relation between the viability percentage of each tumor cell line and extract concentration was analyzed to obtain the IC_50_ (the dose of the drug that reduces survival to 50%) using SigmaPlot 12.0 software [38].

### 4.11. Assessment of Cell Cycle Distribution Using DNA Flow Cytometry

DNA flow cytometry for cell cycle distribution was performed as described by Shati Ali A et al. [39]. In brief, the cells were treated with the IC_50_ of the leaf extract for 48 h, collected by trypsinization, washed with ice-cold PBS, and re-suspended in 0.5 mL of PBS. Ten ml of 70% ice-cold ethanol was gently added while vortexing. The cells were kept at 4 °C for one hour and stored at −20 °C until analysis. Upon analysis, fixed cells were washed and resuspended in 1 ml of PBS containing 50 mg/mL RNase A and 10 mg/mL propidium iodide (PI). After incubating the cells at 37 °C for 20 min, they were analyzed for DNA content by FACSVantage^TM^ (Becton Dickinson Immunocytometry Systems, San Jose, CA, USA). For each sample, 10,000 events were acquired. CELLQuest software, version 7.5.3, from Becton Dickinson Immunocytometry Systems, San Jose, CA, USA was used to calculate the cell cycle distribution. Each treatment was repeated three times, and the data represent the mean ± SD of three replicates.

### 4.12. Apoptosis Analysis

Cells treated with the extract for 48 h were trypsinized and washed twice with PBS. An apoptosis assessment was carried out using the Annexin V-FITC/PI Apoptosis Detection Kit, Cell Signaling Technology (CST), according to the instructions of the manufacturer. In brief, the cells were resuspended in 0.5 mL of binding buffer containing 5 μL of Annexin V-FITC and 5 μL of PI (staining solution). The suspension was incubated at room temperature, with gentle mixing for 15 min, in a dark place. Finally, the cells were subjected to FACS analysis using a Cytek^®^ Northern Lights 2000 spectral flow cytometer (Cytek Biosciences, Fremont, CA, USA) using SpectroFlo™️ Software version 2.2.0.3 (Cytek Biosciences), USA

### 4.13. Statistical Analysis

Data were statistically analyzed using IBM SPSS statistical software, version 21. The values were interpreted as the mean standard deviation (SD) for all the data sets, which were collected in triplicate from more than one biological replicate. To evaluate the difference between the treated and untreated samples, a one-way ANOVA was employed, along with Tukey’s post hoc test. *p* < 0.05 was considered statistically significant.

## 5. Conclusions

In conclusion, the main findings of this study are as follows: (i) This study is the first report of the ethanolic leaf extract of *M. communis* exhibiting strong antibacterial activity against both tuberculosis and non-tuberculosis species of mycobacteria by disrupting the bacterial cell wall. (ii) The extract exhibited biofilm inhibition and eradicated mature and extra-mature biofilms of *M. smegmatis* and *S. aureus*. (iii) This study is the first report of the ethanolic leaf extract of *M. communis* showing anticancer activity against diverse types of cancer cell lines. These findings suggest that *M. communis* leaf extract’s secondary metabolites could be used as therapeutic agents against mycobacterial infections and also be developed to treat breast, liver, cervix, and colon cancer. Further evaluation, active compound isolation, and in vitro and in vivo evaluations are recommended for future research on these active ingredients.

## Figures and Tables

**Figure 1 pharmaceuticals-17-00872-f001:**
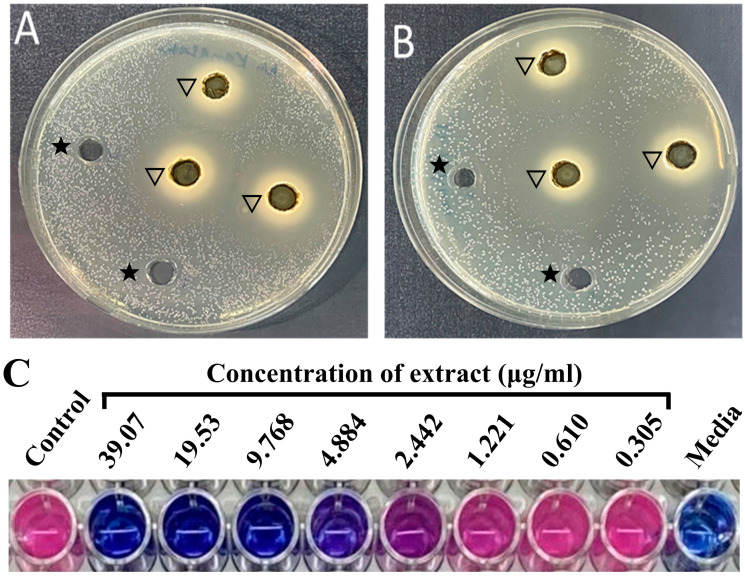
Zone of inhibition and MIC of *M. communis* leaf extract. (**A**) Zone of inhibition of *Mycobacterium kansasii*. (**B**) Zone of inhibition of *Mycobacterium fortuitum* ATCC6841. (**C**) Two-fold dilutions of the extract (µg/mL) used in the culture wells showed an inhibitory effect on the growth of *M. kansasii* ATCC35775 at 4.884 µg/mL. The pink color of the well indicates growth, while the blue color indicates inhibition of the growth of bacteria. ★: well contains ethanol; ▽: well contains extract.

**Figure 2 pharmaceuticals-17-00872-f002:**
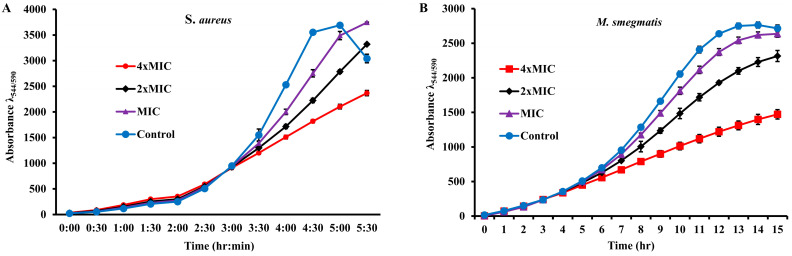
Growth of *S. aureus* and *M. smegmatis* in the presence of the extract: The growth of (**A**) *S. aureus* and (**B**) *M. smegmatis* was monitored in the presence of the MIC, 2× MIC, and 4× MIC of the extract by recording fluorescence readings at 30 min (*S. aureus*) and 1 h (*M. smegmatis*) intervals, respectively, using alarmBlue as a growth indicator.

**Figure 3 pharmaceuticals-17-00872-f003:**
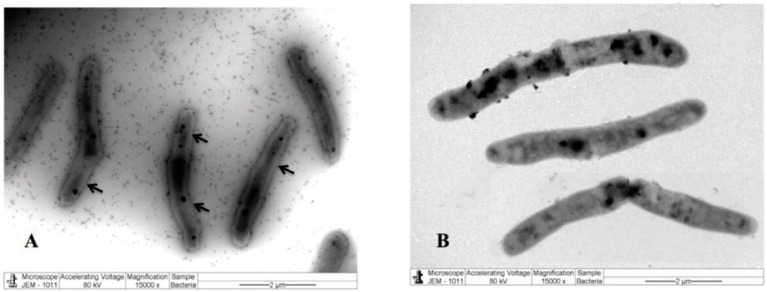
SEM images of *M. smegmatis* cells: Tomographic images of untreated (**A**) and treated (**B**) *M. smegmatis* cells. The arrows show that the cell wall is intact in untreated cells but damaged in treated cells.

**Figure 4 pharmaceuticals-17-00872-f004:**
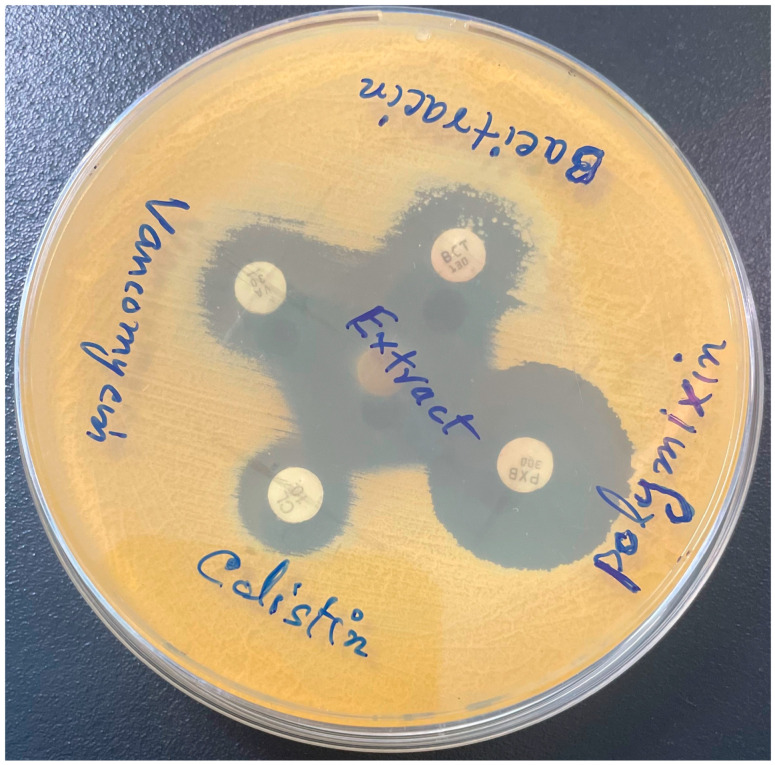
Synergy between antibiotics and extract. Bacitracin and vancomycin exhibited bridging of the zone of inhibition in combination with the extract against *M. smegmatis*.

**Figure 5 pharmaceuticals-17-00872-f005:**
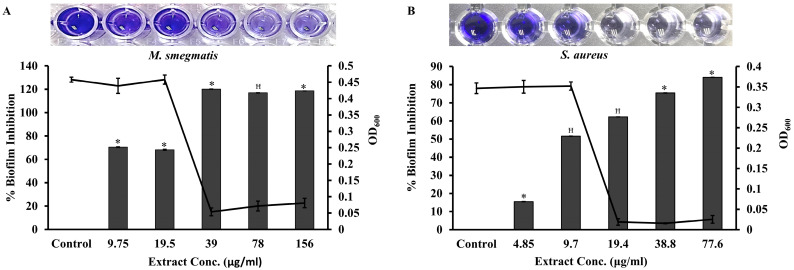
Inhibition of *S. aureus* and *M. smegmatis* biofilm formation by *M. communis* leaf extract. The growth of (**A**) *M. smegmatis* and (**B**) *S. aureus* in the presence of different concentrations of the extract was measured by optical density at 600 nm. The biofilms formed at these concentrations were estimated by crystal violet absorbance at 570 nm. The inlets show the crystal violet-stained biofilms of one of the three replicate wells. *: *p* < 0.002, ꟸ: *p* < 0.006.

**Figure 6 pharmaceuticals-17-00872-f006:**
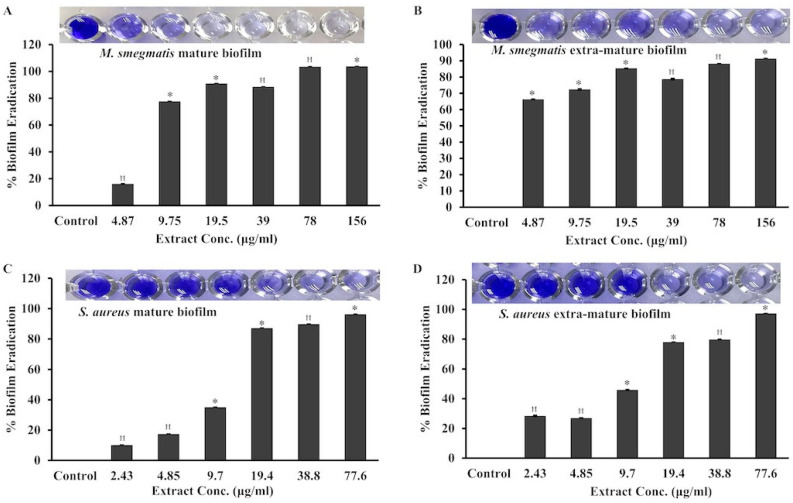
Eradication of mature biofilms by *M. communis* leaf extract. Mature and extra-mature biofilms of *M. smegmatis* (**A**, **B**) and *S. aureus* (**C**, **D**) were subjected to different concentrations of extract treatment. The biofilms that remained after treatment were developed by the crystal violet method and estimated by absorbance at 570 nm. The percentage of biofilms eradicated was calculated and plotted against the extract concentrations. The inlets show the crystal violet-stained biofilms in one of the three replicate wells. *: *p* < 0.002; ꟸ: *p* < 0.006.

**Figure 7 pharmaceuticals-17-00872-f007:**
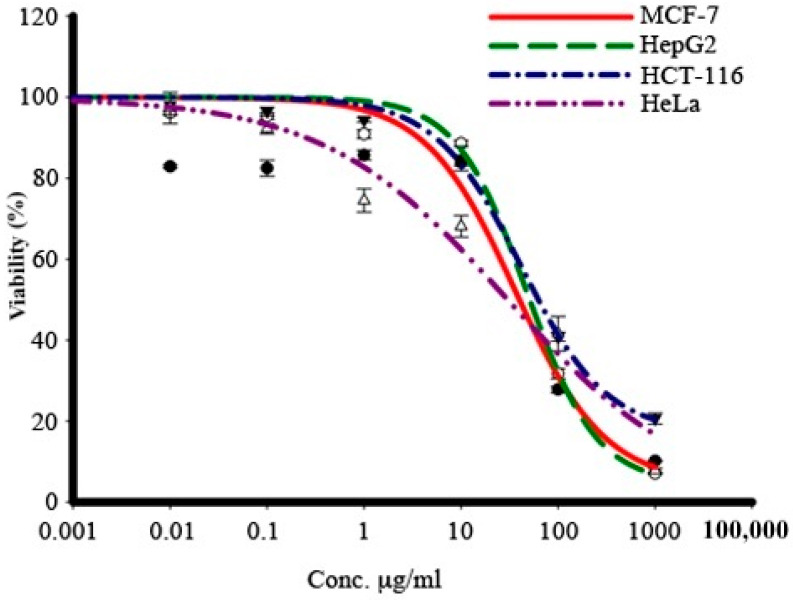
IC_50_ values of the *M. communis* leaf extract against cancer cell lines. The percentage of viable cells after 72 h of treatment with various extract concentrations.

**Figure 8 pharmaceuticals-17-00872-f008:**
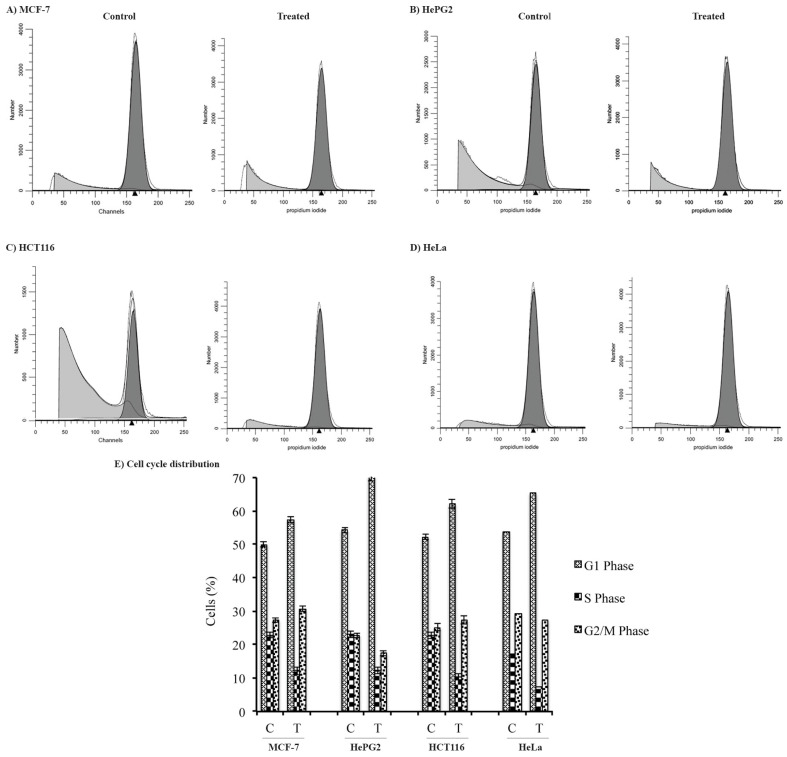
Cell cycle analysis of cancer cell lines after *M. communis* leaf extract treatment. (**A**–**D**): The cell cycle distribution of cells of MCF-7 (**A**), HepG2 (**B**), HCT116 (**C**), and HeLa (**D**) analyzed by DNA flow cytometry after their individual treatment with a plant extract (treated). Similarly, the cell cycle distribution of all cancer cell lines was analyzed after their treatment with DMSO only (control). (**E**) A bar graph showing the percentage of cells for each cell line in different phases of the cell cycle after their treatment with the plant extract (T) or DMSO (C).

**Figure 9 pharmaceuticals-17-00872-f009:**
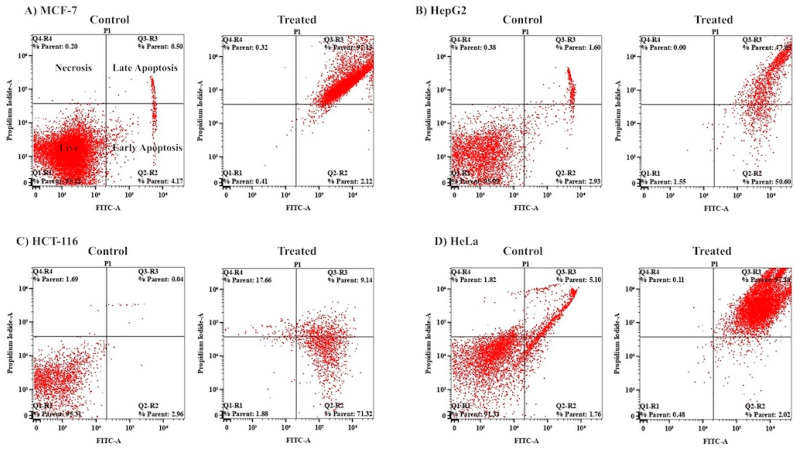
*M. communis* leaf extract induces apoptosis. (**A**–**D**): Scatter plots of flow cytometry analysis showing the distribution of cells of MCF-7 (**A**), HepG2 (**B**), HCT116 (**C**), and HeLa (**D**) into the early and late phases of apoptosis and necrosis after treatment with the plant extract (treated) or DMSO (control) and subsequent staining with Annexin V-FITC/PI. A representation of the four quadrants is given in the scatter plot in the top left corner.

**Table 1 pharmaceuticals-17-00872-t001:** Antimycobacterial activity of *M. communis* leaf extract against pathogenic mycobacterial strains.

Origin	Organism	Zone of Inhibition Mean ± SD (mm)	MIC (μg/mL)	MBC (μg/mL)
SGM	*M. avium ATCC25291*	25.1 ± 4.1	19.53	156.2
	*M. kansasii*	20.3 ± 1.1	9.768	78.14
	*M. kansasii* ATCC35775	25.6 ± 3.2	4.884	39.07
	*M. xenopi*	20.5 ± 2.6	19.53	156.2
RGM	*M. abscessus*	20.6 ± 3.2	39.07	312.5
	*M. fortuitum ATCC6841*	24.3 ± 3.7	156.2	937.2
	*M. mucogenicum*	22.6 ± 3.5	9.768	156.2
MTB	*M. tuberculosis ATCC25177*	26.3 ± 2.5	78.14	625
	*M. tuberculosis RIF-R*	22.6 ± 2.5	312.5	1250

**Table 2 pharmaceuticals-17-00872-t002:** The zones of inhibition of antibiotics alone and in combination with *M. communis* extract.

Antibiotics	Zone Diameter (mm) in Alone	Zone Diameter (mm) in Combination with *M. communis* Leaf Extract
Vancomycin	20 ± 1	26 ± 0
Colistin	16 ± 1	25 ± 2
Bacitracin	16 ± 1	21 ± 0
Polymyxin	25 ± 0	25 ± 0
Chloramphenicol	15 ± 2	24 ± 1
Gentamycin	35 ± 2	24 ± 1
Clarithromycin	31 ± 1	26 ± 2
Ofloxacin	35 ± 2	24 ± 0
Tetracycline	45 ± 2	35 ± 1

**Table 3 pharmaceuticals-17-00872-t003:** Cytotoxic activities (IC_50_) of *M. communis* leaf extract (µg/mL) against different tumor cell lines. Results are expressed as the mean ± SD for three different independent replicates.

Tumor Cell Line	IC_50_ (µg/mL)	Tumor Cell Line	IC_50_ (µg/mL)
MCF-7	41.5 ± 0.6	HCT116	83 ± 2.5
HepG2	53.3 ± 0.6	HeLa	33.3 ± 3.6

**Table 4 pharmaceuticals-17-00872-t004:** Cell cycle distribution of cancer cells after treatment with *M. communis* leaf extract. Results are expressed as the mean ± SD for three independent replicates.

Cancer Cell	Extracts	Cell Phases		
		G_1_	S	G_2_/M
MCF-7	Control	49.92 ± 0.74	22.75 ± 0.8	27.33 ± 0.7
	Treated	57.33 ± 0.9	12.09 ± 1.2	30.58 ± 0.81
HepG2	Control	54.23 ± 0.93	23.06 ± 2.01	22.71 ± 1.7
	Treated	70.52 ± 0.79	12.15 ± 1.02	17.33 ± 0.75
HCT116	Control	52.13 ± 1.52	22.75 ± 0.98	25.12 ± 0.75
	Treated	62.22 ± 0.78	10.38 ± 0.58	27.4 ± 2.02
HeLa	Control	53.69 ± 0.87	17.09 ± 0.98	29.22 ± 1.02
	Treated	65.35 ± 1.35	7.25 ± 0.94	27.4 ± 1.31

G_1_, S, G_2_, and M are the phases of cell cycle.

## Data Availability

The data generated or analyzed during this study are included in this published article.

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
