# Peer review of "Antimycobacterial and Anticancer Properties of Myrtus communis Leaf Extract"

_pharmaceuticals, 2024, doi:10.3390/ph17070872_

Round 1

Reviewer 1 Report (Previous Reviewer 2)

Comments and Suggestions for Authors

After carefully evaluating the clarity and novelty of the information in the resubmitted article, I believe that the manuscript is well-written and the analysis is compelling.

A few minor points need to be addressed:

1. Regarding one of the questions and the response in the previous review:

Question: The authors mention that "This is the first report of M. communis extract showing anticancer activity against diverse types of cancer cell lines.", but there are plenty of other studies in the scientific literature evaluating the antiproliferative activity of M. communis.

Response: This is a fact. There is not any study where M. communis ETHANOLIC LEAF extract has been investigated agaInst all the four cancer cell lines for anti-proliferative activity. However, there are studies where essential oil and other solvent extracts of mostly seeds have been used for such activity.

I understand, but then the authors must rephrase the conclusions in order to be more specific, because the actual version of the manuscript states that "This is the first report of M. communis extract showing anticancer activity against diverse types of cancer cell lines." .

2. Furthermore, the introduction must include more relevant and recent references, considering that in the present version of the manuscript all references cited in the introduction are published before 2021, and some of them are more than 20 years old. 

Comments on the Quality of English Language

Minor editing of English language required

Author Response

Dear Reviewer,

Please see the attached file for responses to your comments and suggestions.

Best,

Reviewer 2 Report (New Reviewer)

Comments and Suggestions for Authors

The manuscript describes the antimycobacterial properties of an ethanolic extract of Myrtle leaves. In addition, the antitumor and antibacterial properties of this extract were assessed. I have no complaints about the methods for assessing biological activity and the quality of the experiment, but there are many questions about the methodology of the entire work.

The authors take air-dry raw materials - it is not clear where and when they were collected. But the composition and content of various substances can very much depend on the place where the raw materials are collected and on the time of collection too. Examples for myrtle can be viewed at the link (Al-Snafi, A.E., Teibo, J.O., Shaheen, H.M. et al. The therapeutic value of Myrtus communis L.: an updated review. Naunyn-Schmiedeberg's Arch Pharmacol (2024). https:// doi.org/10.1007/s00210-024-02958-3). The basic rule of scientific research is that information reported by the authors can be reproduced by other researchers. Based on the data in this manuscript, researchers will collect myrtle leaves, but the inhibition of mycobacterial growth observed by the authors may not occur. Because the composition of biologically active substances is different. It turns out that the value of the information published by the authors is zero. At the same time, modern equipment, in particular GC-MS, makes it possible to determine the main components of the extract at low cost.

An article can only be considered for publication in a sufficiently high-ranking Pharmaceuticals journal if major revisions are made. Namely:

1) The place and time of collection of raw materials must be indicated. Not a reference, but information in its pure form.

2) The main components of the extract must be identified. If they were defined earlier, then this data should still be given at least in the supplementary. And it must be discussed from the point of view of the possible responsibility of individual components for antimycobacterial properties.

3) Quantitative characteristics of the extract, mass fraction of extraction from raw materials must be given

4) The introduction should be rewritten. The listing of well-known truths about the importance of antitumor and antibacterial agents is uninformative and uninteresting. For the essence of this work, it is much more important to know what properties of myrtle extracts have been previously published, what components are responsible for the known activity.

5) Since antimycobacterial properties for this extract have been identified for the first time, in the Discussion section it is necessary to discuss which components may be responsible for this activity. If none of the known components of the extract is an antimycobacterial agent, then the conclusion that a more thorough study of the composition of the extract is necessary will be justified.

6) I would like to note that the observed effects (antimycobacterial, antibacterial and antitumor) are not as strong as the authors describe them. The values of the obtained inhibitory and semi-inhibitory concentrations are quite normal for plant extracts.

Author Response

Dear Reviewer,

Please see the attached file for responses to your comments and suggestions.

Best,

Reviewer 3 Report (New Reviewer)

Comments and Suggestions for Authors

In the manuscript, the authors investigated the antimycobacterial and anticancer properties of Myrtus communis leaf extract. Although I appreciated the assays studied and the presentation of results from these assays, there is a non-negligible gap between the results and potential components in the leaf extract. The lack of information about the components of leaf extract makes the manuscript not possible to publish in the Pharmaceutical Journal. 

The leaf extract has been prepared in ethanol and then dried after filtration. So it is a crude extract. They did not have any phytochemical characterization to determine the components. Because the content of the extract depends on the solvent system used, determining the chemical profile is essential to the comparison to the other studies in the scientific literature evaluating bioactivities of M. communis content. Understanding which natural compound can be attributed to the possible biological activities without the chemical profile of the extract is impossible. 

I strongly recommend that the authors check the plant extraction/isolations printed in the last issues of the pharmaceutical journal to understand the importance of chemical component analyses.

Author Response

Dear Reviewer,

Please see the attached file for responses to your comments and suggestions.

Best,

Round 2

Reviewer 2 Report (New Reviewer)

Comments and Suggestions for Authors

The authors were somewhat deceitful or did not understand the message of reworking the Introduction section. The main conclusion from the article is that by taking the ethanol extract of myrtle leaves, we can fight tuberculosis, cancer and some bacteria. However, the authors should provide information on what other effects we can expect from the ethanol extract of myrtle leaves based on the studies of other authors. But this information is not in the introduction. At the same time, a Google search yields a number of publications on this topic:

https://doi.org/10.1016/j.jams.2011.09.015

DOI: 10.52711/0974-360X.2023.00648

DOI: 10.1177/1934578X0900400616

AlAnbori, Dalia K.A, AlNimer, Marwan S.M, & AlWeheb, Athraa M (2008). Antibacterial activity of ethanolic extract of Myrtus communis L leaves against salivary Mutans streptococci. Saudi Dental Journal, 20(2), 82-87.

DOI: 10.4103/0974-8490.75449

DOI: https://doi.org/10.18502/ijml.v6i3.1404

In addition, I did not find any data in the supplementary on what the weight fraction of the extract is (as a percentage) of the mass of air-dried raw materials.

Author Response

Dear Reviewer,

Please see the point-by-point responses to your comments and suggestions in the attached file.

Best,

Mushtaq

Reviewer 3 Report (New Reviewer)

Comments and Suggestions for Authors

 The manuscript has been sufficiently improved to warrant publication in Pharmaceuticals

Author Response

We thank the reviewer for his or her satisfaction with the quality and comprehensiveness of the manuscript.

Regards,

Mushtaq A. Mir

This manuscript is a resubmission of an earlier submission. The following is a list of the peer review reports and author responses from that submission.

Round 1

Reviewer 1 Report

Comments and Suggestions for Authors

After a critic revision in my opinion whole article needs to be re-evaluated and integrated with some others experiments by the authors.  

The abstract needs to be reformulated as it is too schematic.

The introduction lacks organicity, and the connection between what is reported in the literature and the aim of the work is not well understood. 

From line 51 to 55 keep attention to the font 

line 55, 60, 68 when you cited for the first time the name of the microorganism need to be reported in the long form 

line 63 you can write only the acronym 

line 76 keep attention to the correct verb form

from line 76 to 86 and from 94 to 98 need the bibliography 

line 112 not correct bibliography 

Regarding the section 2.1 you reported the extraction procedure previously described by you; 

On my experiences the characterization of the extract is partial. In fact you report a soxhlet extraction which is not appropriate for the subsequent characterization of the volatiles at GC-MS which with the soxhlet temperatures and the fact that it is an open system in itself is not suitable for terpenes (that you reported) and therefore certainly does not represent the real composition of your extract. 

This does not even allow us to reason about which natural compound can be attributed for the possible biological activity. For this reason I advise the authors to proceed with an exhaustive characterization of the ethanolic extract used for the biological experiments.

At this time the article can not be consider for publication.

Comments on the Quality of English Language

I don't have comments about the quality of English 

Author Response

Query: The abstract needs to be reformulated as it is too schematic.

Response; The abstract is the sum up of the study, I believe it is perfect according to the results and outcomes of the study. 

Query: The introduction lacks organicity, and the connection between what is reported in the literature and the aim of the work is not well understood. 

Response: the introduction is well written with the aim of the study and its implications.

Query: From line 51 to 55 keep attention to the font 

line 55, 60, 68 when you cited for the first time the name of the microorganism need to be reported in the long form 

Response: This has been fixed.

Query: line 63 you can write only the acronym 

Response: This has been followed.

Query: line 76 keep attention to the correct verb form

Response: this has been followed

Query: from line 76 to 86 and from 94 to 98 need the bibliography

Response: references have been incorporated

 Query: line 112 not correct bibliography 

Response: This has been addressed

Query: Regarding the section 2.1 you reported the extraction procedure previously described by you; 

On my experiences the characterization of the extract is partial. In fact you report a soxhlet extraction which is not appropriate for the subsequent characterization of the volatiles at GC-MS which with the soxhlet temperatures and the fact that it is an open system in itself is not suitable for terpenes (that you reported) and therefore certainly does not represent the real composition of your extract. 

This does not even allow us to reason about which natural compound can be attributed for the possible biological activity. For this reason I advise the authors to proceed with an exhaustive characterization of the ethanolic extract used for the biological experiments.

Response: I think the reviewer hasn't seen articles where soxhlet extraction method has been used. In soxhlet method, the ethanol is boiled and after condensation it falls in drops on the substrate (leaf powder), so there is no question of high temperature. Whatever compounds are extracted in ethanol by this method, they do exhibit the antibacterial and anticancer activities (as shown in this manuscript).  Moreover, in this manuscript leaf extract's effects are not attributed to any specific compound. The GC-MS analysis identified the compounds published in our previous article, and the present study showed its biological properties. We do not claim of any specific compound that is responsible for such properties, though there are compounds which have been shown to posses such activities. The purification of the compounds and the validation of these biological properties is our future work. 

Reviewer 2 Report

Comments and Suggestions for Authors

After carefully reviewing the present study, I believe that the current form of the manuscript should be rejected, and the authors can be encouraged to resubmit the manuscript only after updating it. The main concern are:

First of all, all the figures and tables are missing, thus a correct assessment of the results of this study cannot be made.

The authors used a strain of S. aureus to perform some analysis, although the purpose of the study was to evaluate antimycobacterial and anticancer activity of the extract

The authors mention that "This is the first report of M. communis extract showing anticancer activity against diverse types of cancer cell lines.", but there are plenty of other studies in the scientific literature evaluating the antiproliferative activity of M. communis.

The manuscript needs extensive English language revisions.

Comments on the Quality of English Language

The manuscript needs extensive English language revisions.

Author Response

Query: First of all, all the figures and tables are missing, thus a correct assessment of the results of this study cannot be made.

Response; I apologize, though all tables and figures are there.  

Query; The authors used a strain of S. aureus to perform some analysis, although the purpose of the study was to evaluate antimycobacterial and anticancer activity of the extract

Response: Since we studied the biofilm formation of M. smegmatis first time in our laboratory. Therefore, for control we took S. aureus and for which we knew it makes strong biofilms and the extract inhibits its biofilm formation.  This was the only reason for performing few experiments on S. aureus. 

Query; The authors mention that "This is the first report of M. communis extract showing anticancer activity against diverse types of cancer cell lines.", but there are plenty of other studies in the scientific literature evaluating the antiproliferative activity of M. communis.

Response: This is a fact. There is not any study where M. communis ETHANOLIC LEAF extract has been investigated agaInst all the four cancer cell lines for anti-proliferative activity. However, there are studies where essential oil and other solvent extracts of mostly seeds have been used for such activity.